# (Im)possibility of Automated Hallucination Detection in Large Language Models

**Amin Karbasi**
Yale University
amin.karbasi@yale.edu

**Omar Montasser**
Yale University
omar.montasser@yale.edu

**John Sous**
Yale University
john.sous@yale.edu

**Grigoris Velegkas**
Yale University
grigoris.velegkas@yale.edu

## Abstract

Is automated hallucination detection fundamentally possible? In this paper, we introduce a theoretical framework to rigorously study the (im)possibility of automatically detecting hallucinations produced by large language models (LLMs). Our model builds on the classical Gold-Angluin framework of language identification (Gold, 1967; Angluin, 1980) and its recent adaptation by Kleinberg & Mullainathan (2024) to the language generation setting. Concretely, we investigate whether an algorithm—trained on examples from an unknown target language $K$, chosen from a countable collection of languages $\mathcal{L}$, and given access to an LLM—can reliably determine if the LLM's outputs are correct or constitute hallucinations.

First, we establish a strong equivalence between hallucination detection and the classical problem of language identification. Specifically, we prove that any algorithm capable of identifying languages (in the limit) can be efficiently transformed into one that reliably detects hallucinations, and conversely, successful hallucination detection strategy inherently implies language identification. Given the notorious difficulty of language identification, our first result implies that hallucination detection is *impossible* for most collections of languages.

Second, we show that once we enrich the detector's training data, i.e., providing it with both positive examples (correct statements) and negative examples (explicitly labeled incorrect statements)— the conclusion dramatically changes. Under this enriched training regime, we show that automated hallucination detection is *possible* for any countable collection $\mathcal{L}$.

Our theoretical results, thus, underscore the fundamental importance of expert-labeled feedback in the practical deployment of hallucination detection methods, reinforcing why feedback-based approaches, such as reinforcement learning with human feedback (RLHF), have proven so crucial in improving the reliability and safety of real-world LLMs.

## 1 Introduction

The recent breakthroughs in Large Language Models (LLMs) have significantly advanced the state-of-the-art in natural language processing and broader machine learning tasks (OpenAI et al., 2024; Team et al., 2024). Contemporary models routinely demonstrate exceptional performance across diverse tasks, including mathematical reasoning, complex problem-solving, and generating coherent, contextually appropriate text (Bubeck et al., 2023; Touvron et al., 2023).

However, alongside these remarkable capabilities, a critical limitation has emerged: LLMs frequently produce *hallucinations*—outputs that appear fluent and convincing yet are factually incorrect (Ji et al., 2023). Hallucinations significantly limit the trustworthiness of LLMs,

posing substantial risks when deploying them in sensitive applications, and raising urgent concerns around ethics, reliability, and societal impacts (Weidinger et al., 2021; Zhang et al., 2023; Azamfirei et al., 2023).

A promising approach to addressing hallucinations is the development of automated detection mechanisms. Unfortunately, practical attempts to detect hallucinations using LLMs themselves as detectors have faced limitations. Empirical studies indicate that LLMs perform significantly worse than humans at identifying hallucinations, and typically require reliable external feedback—such as explicit labeling by experts—to improve (Kamoi et al., 2024a;b). Despite these empirical observations, a theoretical understanding of these practical difficulties has remained open:

*Is automated hallucination detection inherently difficult, or can we expect it to become easier as models improve?*

To address this gap, we introduce a formal theoretical framework inspired by classical learning theory—particularly the foundational work of Gold and Angluin on *language identification* (Gold, 1967; Angluin, 1979; 1980), and its recent adaptation to the context of *language generation* by Kleinberg & Mullainathan (2024). Specifically, we propose a novel theoretical abstraction to formally study the feasibility of reliably detecting hallucinations produced by language models. In our model, the hallucination detector is provided with a corpus of training data coming from some unknown target language $K$ and is allowed to interact with an LLM, whose outputs we denote by the set $G$. Conceptually, the language $K$ encodes all statements that are factually correct, while any output outside of $K$ is considered a *hallucination*. We say that a hallucination detection algorithm is *successful* if, after observing sufficiently many examples from $K$ and interacting extensively with the LLM, it eventually determines correctly whether or not the LLM produces hallucinations. Formally, this means that if $G \subseteq K$, the detector should eventually conclude that the LLM does not hallucinate, whereas if $G \not\subseteq K$, meaning the LLM generates outputs outside of $K$, the detector should correctly identify that the LLM hallucinates.

Our first main result formally establishes an equivalence between hallucination detection and the classical problem of language identification, which is known to be inherently challenging (Gold, 1967; Angluin, 1980). The practical implication is summarized concisely below:

> **Informal Result I.** Automated detection of hallucinations by a detector that is trained only on correct examples (positive examples) is inherently difficult and typically impossible without additional assumptions or signals.

Thus, this result provides theoretical justification for the challenges encountered in practice when trying to automatically detect whether an LLM hallucinates.

Given this negative finding, we next examine a more optimistic scenario, inspired both by classical theory (Gold, 1967) and modern empirical approaches (Kamoi et al., 2024a), in which the detector receives both correct statements (*positive examples*) and explicitly labeled incorrect statements (*negative examples*). Under these conditions, the outlook changes dramatically:

> **Informal Result II.** Reliable automated hallucination detection is achievable when the detector is trained using both correct (positive) and explicitly labeled incorrect (negative) examples.

This result has an interesting implication for practical attempts to create hallucination detectors: *explicit expert feedback, particularly negative examples, is critical and fundamentally necessary for automated hallucination detection to succeed.*

**Implications for LLM practice.**   Our framework shows that, with access only to positive examples from the target language, reliable automated hallucination detection is impossible

for broad classes of collections; by contrast, providing explicitly negative examples makes detection feasible. This mirrors empirical findings on LLM-based verifiers (Manakul et al., 2023; Azaria & Mitchell, 2023; Kamoi et al., 2024b;a; Tyen et al., 2023; Niu et al., 2023; Sriramanan et al., 2024; Ji et al., 2023; Zhang et al., 2023), which repeatedly observe large gains when verifiers can consult external sources (e.g., RAGs). These retrieval-augmented tools as the "negative examples" we require in our positive result. While this analogy is imperfect (real retrieval is incomplete and noisy), the practical takeaway is clear: effective detectors need access to falsification signals, not just additional positive text.

## 1.1 Related Work

### 1.1.1 Theoretical frameworks for LLMs

Our proposed framework builds on seminal works in learning theory, including the seminal Gold-Angluin framework (Gold, 1967; Angluin, 1979; 1980) for language identification, and its recent adaptation to language generation by Kleinberg & Mullainathan (2024). Following Kleinberg's and Mullainathan's formulation, Li et al. (2024) extended this perspective, using a learning-theoretic lens to characterize when "uniform" and "non-uniform" language generation are achievable. Further, recent works by Kalavasis et al. (2025; 2024); Charikar & Pabbaraju (2024) explored notions of generation "with breadth," demonstrating that this goal is inherently harder than standard language generation, and, in some cases, as challenging as language identification itself. In a similar spirit, Peale et al. (2025) formalized and studied a notion of "representative generation," and showed that it is possible to achieve it in the limit for all countable collections of languages. In a complementary direction, Raman & Raman (2025) studied language generation from *noisy* examples, considering scenarios where training data includes instances outside the target language $K$. In a concurrent and independent work, Kleinberg & Wei (2025) studied a hallucination-breadth trade-off based on a notion of *density* of languages. Subsequent work showed that generation need not be closed under finite unions (Hanneke et al., 2025). Diverging from the Gold-Angluin framework, Kalai & Vempala (2024) connected *calibration* in generation to increased hallucination rates. Additionally, recent analyses by Peng et al. (2024); Chen et al. (2024) identified fundamental limitations of transformer architectures. Using techniques from communication complexity, they proved transformers are incapable of composing functions when domains become sufficiently large, providing rigorous evidence for inherent hallucination tendencies in LLMs, given that function composition underlies reasoning (Guan et al., 2024). Lastly, Xu et al. (2024) leveraged complexity theory tools to demonstrate inevitable hallucinations in LLMs under certain assumptions. Earlier work by Hanneke et al. (2018) also illustrates the value of using external feedback to mitigate hallucinations of generative models. Our work contributes to this literature which aims to give theoretical insights into the capabilities and limitations of LLMs.

### 1.1.2 Empirical works on automated hallucination detection

Automated hallucination detection has recently gained significant attention, driven by the practical urgency to mitigate hallucinations. Several empirical approaches have emerged to tackle this challenge. For instance, Manakul et al. (2023) introduce SelfCheckGPT, a black-box hallucination detection method that relies solely on stochastic sampling of model responses. The core intuition of their method is that factually accurate responses are typically consistent and frequent, whereas hallucinated outputs tend to vary and contradict each other. In contrast to the black-box consistency-based method, Azaria & Mitchell (2023) propose leveraging the internal hidden states of the LLM to classify outputs as hallucinated or factual. Notably, their classifier is trained using an explicitly labeled dataset comprising sentences marked as either correct or incorrect, highlighting the benefit of explicitly supervised hallucination detection. Their results significantly outperform probability-distribution-based methods, illustrating the advantage of internal-state supervision and leveraging annotated datasets to perform this task. Building upon these empirical insights, Kamoi et al. (2024a) conduct a comprehensive evaluation demonstrating the limitations of current LLM-based hallucination detection approaches. In particular, they show that LLMs perform poorly as detectors when evaluating responses generated by other models, emphasizing

the challenge in using LLMs for automated hallucination detection without robust external signals. Echoing this observation, Tyen et al. (2023) further demonstrate that introducing even minimal human feedback greatly enhances the capability of LLMs to reliably detect hallucinations. Similarly motivated, Niu et al. (2023) illustrate the benefits of fine-tuning LLMs using carefully curated, high-quality labeled datasets containing explicit annotations of hallucinations. This supervised fine-tuning improves hallucination detection performance and underscores the importance of explicitly labeled negative examples. In another related work Sriramanan et al. (2024), study computational efficient methods for automated hallucination methods in several different cases, including black-box access to the model, white-box access to the model, and access to external databases for validation. Similarly, Rawte et al. (2024) study the usefulness of such databases in detecting hallucinations. For comprehensive surveys on the broad topic of hallucinations in LLMs, including various detection methods discussed above, we refer the interested reader to Ji et al. (2023); Zhang et al. (2023).

Our theoretical findings provide formal validation for these empirical results, clearly highlighting the crucial role played by explicitly labeled negative examples in successful hallucination detection.

## 2 Model and Formal Results

### 2.1 Model

In this section we define the formal model we consider in this work. We denote by $\mathcal{L} = \{L_1, L_2, \ldots\}$ a countable collection of candidate languages, where each language $L_i$ is a subset of some countable domain $\mathcal{X}$. We assume that we have membership access to the collection $\mathcal{L}$, meaning that for any $i \in \mathbb{N}$ and $x \in \mathcal{X}$ we can ask whether $x \in L_i$. We allow $\mathcal{L}$ to contain multiple occurrences of the same language, *i.e.* there might exist $i \neq j$ such that $L_i = L_j$.[1] Each language $L_i$ can have finite or infinite cardinality.[2] We define an enumeration of a language $L$ to be an infinite sequence $E = (w_1, w_2, w_3, \ldots)$ such that for all $i \in \mathbb{N}$ we have $w_i \in L$, and for all $x \in L$ there is some $j \in \mathbb{N}$ such that $w_j = x$. Notice that this allows for repetitions of strings, but, crucially, for any given string $x \in L$ there is a finite index where this string appears.

We define the hallucination detection game as the following interaction between a learner and an adversary: the adversary picks a target language $K \in \mathcal{L}$, an arbitrary enumeration $E = (w_1, w_2, \ldots)$ of $K$, and a target set $G \subseteq \mathcal{X}$. We say that $G$ *hallucinates* with respect to $K$ if it contains elements outside of $K$, *i.e.*, if $G \not\subseteq K$.[3] We denote by $E_t = (w_1, \ldots, w_t)$ the prefix of the first $t$ elements of $E$. In every timestep $t = 1, 2, \ldots$, the learner observes $w_t$, asks *finitely* many membership queries to $G$, *i.e.*, for finitely many $x_1, \ldots, x_k \in \mathcal{X}$ it can ask if $x_i \in G$, and get the correct response. Then, it has to output its guess $g_t \in \{0, 1\}$ whether $G \subseteq K$; it outputs 0 if it believes $G$ hallucinates and 1 otherwise. We say that the learner detects hallucinations in the limit if for every target language $K \in \mathcal{L}$, enumeration $E$ of $K$, and candidate $G \subseteq \mathcal{X}$ it holds that after sufficiently long but finite $t$ the guesses of the learner become correct, *i.e.*, there exists some $t_0 \in \mathbb{N}$ such that $g_t = \mathbb{1}\{G \subseteq K\}, \forall t \geq t_0$. The formal definition is provided below.

**Definition 1** (Hallucination Detection in the Limit). *Fix some $K$ from the language collection $\mathcal{L} = \{L_1, L_2, \ldots\}$ and some set $G \subseteq \mathcal{X}$. The hallucination detection algorithm $\mathcal{D} = (\mathcal{D}_t)$ detects hallucinations for $G$ in the limit if there is some $t^* \in \mathbb{N}$ such that for all steps $t > t^*$, the detector's guess $d_t$ satisfies $d_t = \mathbb{1}\{G \subseteq K\}$. The language collection $\mathcal{L}$ allows for hallucination detection in the limit if there is a hallucination detector that detects in the limit for any $K \in \mathcal{L}$, for any $G \subseteq \mathcal{X}$, and for any enumeration $E$ of $K$.*

---

[1]This is because there might be different canonical representations of the same language.
[2]In the model of Kleinberg & Mullainathan (2024) the languages need to have infinite cardinality; this is not needed in our model.
[3]In the terminology of Bang et al. (2025), this type of hallucination is referred to as "factuality."

To gain some intuition about this model, it is useful to consider a simple example.

**Example 1.** Let $\mathcal{X} = \{0,1\}^*$, *i.e.*, the set of all binary strings of finite length, and $\mathcal{L} = \{L_1, L_2, L_3, \ldots\}$, where $L_i = \{s : |s| \geq i\}$, *i.e.*, the $i$-th language contains all the strings that have length at least $i$. Notice that every language in this collection is regular. Assume the adversary chooses $K = L_2$, *i.e.*, the language of all strings of size at least 2. Then, it has to present to the learner all these strings, one at a time, potentially allowing for duplicates in the presentation. Crucially, for every string $s$ with $|w| \geq 2$, there is some timestep $t_j \in \mathbb{N}$ such that $w_{t_j} = s$. Consider two potential choices of $G$ : the first choice is $G_1 = L_3$ and the second choice is $G_2 = L_1$.[4] In the first case, a successful hallucination detection algorithm should claim, in the limit, that $G_1$ does not hallucinate with respect to $K$, whereas in the second case it should claim that $G_2$ does hallucinate with respect to $K$. To give the reader a first glance of the difficulty of the hallucination detection task, while we have not stated our main result yet (Theorem 2.1), it is worth pointing out that this result, along with Angluin's characterization of language detection in the limit (Theorem A.1), implies that no hallucination detection algorithm exists for this collection.

**Identification and generation in the limit.** Our model is closely related to the Gold-Angluin language identification setting (Gold, 1967; Angluin, 1979; 1980), and the language generation setting of Kleinberg & Mullainathan (2024). In both of these models there is a infinite game between a learner and an adversary: the adversary picks a target $K \in \mathcal{L}$ and an enumeration of that target; however, unlike these models, the adversary does not pick a target set $G \subseteq \mathcal{X}$ as happens in our setting. Similar to these models, in every $t \in \mathbb{N}$ the learner observes a new element from the enumeration. In the identification setting, the goal of the learner is to find an index of the target language $K$ for all but finitely many steps, and in the generation setting the goal is to output *unseen* elements of $K$ for all but finitely many steps. We present a more formal treatment of these settings in Appendix A. Angluin (1980) exactly characterized when identification in this setting is achievable. Her result is largely viewed as an impossibility result, since a very limited number of collections satisfy it. On the other hand, Kleinberg & Mullainathan (2024) showed that the landscape of generation is vastly different: it is achievable for all such collections.

**Connections of theoretical model to practical LLM training.** At this point, it is instructive to pause and consider some common features of these three models; we believe that while they are mathematical abstractions of the practical LLM training process, they capture a lot of important aspects of this process. The way to interpret the different languages of the collection $\mathcal{L}$ is that they capture different "worlds" and the different elements of $\mathcal{X}$ are different "statements." Therefore, each "world" defines precisely which "statements" are accurate and which ones are inaccurate. Admittedly, real-world applications might be more nuanced than that and there could be statements that cannot be easily categorized into accurate or inaccurate ones. Since our model gives a clear taxonomy, it follows that a negative result here can be viewed as a strong indication that in real-world applications hallucination detection is even more challenging.

In our model, we consider an *adversarial enumeration* instead of placing distributional assumptions on the language generation process and the way the outputs of the LLM are generated. While this might look like a restriction of our model at first glance, it turns out that our results carry over to a setting where the data are generated probabilistically; this follows from techniques similar to Kalavasis et al. (2025). We choose to focus on the adversarial setting, following the work of Kleinberg & Mullainathan (2024), to make our exposition easier to follow.

Moreover, in all these models the "learner" is given access only to *positive examples* in the form of elements that belong to the target language. This assumption is capturing the pre-training process of modern machine learning architectures that are trained on a large corpus of datapoints that are elements of the target language and are deployed to act at

---

[4]We underline that we do not place the restriction $G \in \mathcal{L}$, this is only done to illustrate our example.

automatic language identifiers, generators or detectors. We also ignore the fact that the training dataset might be corrupted. Again, this simplification is made to ensure that our negative result reflects an innate difficulty of the hallucination detection task and is not an artifact of inaccuracies contained in the training data.

Next, notice that in all three settings we have discussed so far – language identification, language generation, and hallucination detection – the algorithm never receives feedback about its guesses. This is also largely consistent with the pre-training phase of the LLM training pipeline.

Furthermore, we do not place any computational restrictions on the learning algorithm or the architecture that it relies upon. In fact, we only wish for the detection property to hold "in the limit." This simplification is again made to ensure that any negative results in the setting reveal inherent difficulties of the underlying task and are not mere limitations of the current technologies or computational resources that might be rectified in the future.

Lastly, we underline that throughout our work we consider a *promptless* generation setting. Intuitively, this is also a simplification of the behavior of real-world LLMs, thus negative results in our model should also carry over to applied settings. We emphasize that all these assumptions are largely consistent with prior work on theoretical capabilities and limitations of LLMs (Kalai & Vempala, 2024; Xu et al., 2024; Kleinberg & Mullainathan, 2024; Kalavasis et al., 2024; Charikar & Pabbaraju, 2024; Kalavasis et al., 2025). We believe that our results can be extended to the prompted setting of Kleinberg & Mullainathan (2024), and we leave this as an interesting future direction.

## 2.2 Formal Results

Given the similarities of the different tasks we have described so far—language identification, language generation, and hallucination detection—it is natural to ask: is hallucination detection as easy as generation, as hard as identification, or does it lie somewhere in between? Our first main result gives a precise answer to this question; we show that hallucination detection is as hard as identification.

**Theorem 2.1.** *A countable collection of languages $\mathcal{L} = \{L_1, L_2, \ldots\}$ over some countable domain $\mathcal{X}$ admits an algorithm that detects hallucinations in the limit if and only if $\mathcal{L}$ is identifiable in the limit.*

Given our result and Angluin's characterization (Angluin, 1980) which we state in Theorem A.1, we get the following immediate corollary.

**Corollary 2.2.** *A countable collection of languages $\mathcal{L} = \{L_1, L_2, \ldots\}$ over some countable domain $\mathcal{X}$ admits an algorithm that detects hallucinations in the limit if and only if $\mathcal{L}$ satisfies Angluin's condition (Definition 4).*

Given this largely negative result about the ability to perform automated hallucination detection of LLMs, we next ask what more information is needed by the learner to perform this task. Inspired by Gold's work (Gold, 1967), we consider a modified game termed hallucination detection with negative examples. The main difference is that instead of presenting an enumeration of the target language $K$, the adversary presents an enumeration of the whole domain $\mathcal{X}$ along with a label in $\{0, 1\}$ indicating whether the enumerated element is in the target language or not. We call this type of enumeration a *labeled* enumeration. In stark contrast to our previous result, we show that hallucination detection with negative examples is always possible. The formal description of this game igiven below.

**Definition 2** (Hallucination Detection with Negative Examples in the Limit). *Fix some $K$ from the language collection $\mathcal{L} = \{L_1, L_2, \ldots\}$ and some set $G \subseteq \mathcal{X}$. The hallucination detection algorithm $\mathcal{D} = (\mathcal{D}_t)$ detects hallucinations for $G$ given negative examples in the limit if there is some $t^* \in \mathbb{N}$ such that for all steps $t > t^*$, the detector's guess $d_t$ satisfies $d_t = \mathbb{1}\{G \subseteq K\}$. The language collection $\mathcal{L}$ allows for hallucination detection with negative examples in the limit if there is a hallucination detector that detects in the limit for any $K \in \mathcal{L}$, for any $G \subseteq \mathcal{X}$, and for any labeled enumeration $E$ of $\mathcal{X}$ with respect to the target language $K$.*

Having explained the mathematical setting, we are now ready to state our formal result.

**Theorem 2.3.** *Every countable collection of languages $\mathcal{L} = \{L_1, L_2, \ldots\}$ over some countable domain $\mathcal{X}$ admits an algorithm that, given negative examples, detects hallucinations in the limit.*

## 3 Overview of the Approach

Having discussed our formal setting and results, we now describe the main steps of our technical approach. We start with Theorem 2.1.

### 3.1 Proof of Theorem 2.1

Our approach here is divided into two main steps. First, we show that we can transform any algorithm that achieves identification in the limit in this setting to an algorithm that detects hallucinations in the limit.

**Language identification $\implies$ hallucination detection.** The formal statement of this result is given below.

**Lemma 3.1.** *Let $\mathcal{L}$ be a countable collection of languages over a domain $\mathcal{X}$ that is identifiable in the limit. Then, $\mathcal{L}$ admits an algorithm that achieves hallucination detection in the limit.*

Let us now explain the idea of our approach, which utilizes the identification algorithm in a black-box way. In every timestep $t$, the learner feeds the element $w_t$ the adversary enumerates to the identification algorithm. The identification property (Definition 3) immediately shows that there exists some $t^* \in \mathbb{N}$ (which depends on the choice of the target language $K$ and the enumeration $E$) such that for all $t > t^*$ the identifier's guess $i_t$ satisfies $i_t = i_{t-1}$ and $L_{i_t} = K$. The learner next considers an enumeration of the domain $\mathcal{X} = \{x_1, x_2, \ldots\}$. In the $t$-th step of the execution, the learner uses the membership oracle to check which of the elements $x_1, \ldots, x_t$ belong to the language $L_{i_t}$. Subsequently, the learner also queries the target LLM, modeled as the set $G$, to see which of these elements belong to it. If for all $x_i \in G$ it holds that $x_i \in L_{i_t}$, then the guess of the hallucination detection algorithm for this step will be that the LLM does not hallucinate. We present a formal overview of our hallucination detection strategy in Algorithm 1. We now give the formal proof of our result.

---

**Algorithm 1** Hallucination Detection from Language Identification

---

**Input:** Identification algorithm $I$; enumeration $E = (w_1, w_2, \ldots)$ of $K$; language collection $\mathcal{L}$ (with membership oracle); domain $\mathcal{X}$; LLM output set $G$

1: **for** $t = 1, 2, \ldots$ **do**
2:     Feed $E_t = (w_1, \ldots, w_t)$ to $I$ to obtain guess $i_t$.
3:     Let $\widehat{K} \leftarrow L_{i_t}$.
4:     Enumerate domain prefix $\mathcal{X}_t = \{x_1, \ldots, x_t\}$.
5:     **for** each $x \in \mathcal{X}_t$ **do**
6:         **if** $x \in G$ and $x \notin \widehat{K}$ **then**
7:             **return** $G$ hallucinates
8:         **end if**
9:     **end for**
10:     **return** $G$ does not hallucinate
11: **end for**

---

*Proof of Lemma 3.1.* First, notice that by definition of the identification property, it holds that there exists some $t^* \in \mathbb{N}$ (that depends both on the target language and the enumeration) such that for all $t \geq t^*$ the output of the identifier satisfies $L_{i_t} = K$. Let us now consider any $t > t^*$. We divide our analysis into two disjoint cases, which jointly cover all possible outcomes. First, let us consider the case $G \subseteq K$. Then, for all $t \geq t^*$ we have that if $x_i \in G$ then $x_i \in K$, for all $1 \leq i \leq t$. Thus, our algorithm will correctly claim that the LLM does not

hallucinate for all $t \geq t^*$. Next, we consider the slightly more challenging case $G \nsubseteq K$. By definition, there exists some $x \in \mathcal{X}$ such that $x \in G$ and $x \notin K$. Let $i^* \in \mathbb{N}$ be the smallest index in the enumeration of $\mathcal{X}$ for which this holds, *i.e.*, $x_{i^*} \in G, x_{i^*} \notin K$. Then, for any $t \geq \max\{t^*, i^*\}$ when we consider the prefix of the enumeration $x_1, \ldots, x_t$ the element $x_{i^*}$ will be included in this enumeration. Moreover, it holds that $L_{i_t} = K$. Thus, when the hallucination detector tests the element $x_{i^*}$, it will see that $x_{i^*} \in G$ and $x_{i^*} \notin K$ and it will correctly deduce that the LLM $G$ hallucinates. These two arguments conclude the proof. $\square$

**Hallucination detection $\implies$ language identification.** We now shift our attention to the more technically intricate result which shows that language identification is not harder than hallucination detection. This is also a black-box transformation; it takes as input any hallucination detection algorithm and it constructs an identification algorithm.

**Lemma 3.2.** *Let $\mathcal{L}$ be a countable collection of languages over a domain $\mathcal{X}$ that admits an algorithm that achieves hallucination detection in the limit. Then, $\mathcal{L}$ is identifiable in the limit.*

Before explaining our construction, it is instructive to build some intuition about the difficulty of the language identification task. A natural attempt to achieve language identification is to keep track of all the language that are "consistent" with the current set of examples $E_t$ the adversary has enumerated, that is the set $\mathcal{C}_t = \{L \in \mathcal{L} : E_t \subseteq L\}$. It is not very hard to see that for any language $L_i$ that is not a (strict) superset of the target language $K$, there is some timestep $t_i^*$ such that $L_i \notin \mathcal{C}_t$. Indeed, since $L_i \nsupseteq K$, there exists some $x_i$ which satisfies $x_i \in K, x_i \notin L_i$. Thus, when the adversary enumerates $x_i$ the algorithm will deduce that $L_i \notin \mathcal{C}_t$. What happens if $L_i$ is a (strict) superset of $K$? Unfortunately, in this case the language $L_i$ will always remain consistent with the sample $E_t$. Thus, the strategy of keeping track of the consistent languages is not sufficient to guarantee identification in the limit. Indeed, consider Example 1: no matter what the choice the target language $K$ and the enumeration $E$ of the adversary is, the language $L_1$ will always be consistent with the sample $E_t$. Thus, the consistency-based approach is not sufficient to distinguish between $L_j, j \neq 1$, and $L_1$. Is there a more sophisticated approach that can overcome this obstacle? The seminal result of Angluin (1980) shows that, unless $\mathcal{L}$ satisfies some very strong structural conditions (Definition 4), the answer is largely negative.

The previous discussion highlights that in order to achieve identification in the limit we need to leverage the hallucination detection algorithm to distinguish between languages $L_i$ with $L_i \supsetneq K$ and the target language $K$. Our main insight is that the "consistency-based" approach and the hallucination detection algorithm work in a complementary way: the former allows us to discard languages that are not (strict) supersets of $K$, while the latter helps us rule out languages that are (strict) supersets of $K$.[5] Neither of these approaches is sufficient to be used for language identification on its own, but it turns out that a carefully crafted approach that coordinates their behavior gives the desired result.

We now explain our algorithm in more detail; its formal description is given in Algorithm 2. In each step $t$ we create the set $\mathcal{C}'_t = \{L_i \in \mathcal{L} : E_t \subseteq L_i, 1 \leq i \leq t\}$, *i.e.*, the set of the languages whose index is at most $t$ and are consistent with the elements $E_t$ that have been enumerated so far. Notice that this can be achieved with finitely many queries to the membership oracle for $\mathcal{L}$.[6] Let $L_{i_1}, \ldots, L_{i_k}$ be the languages of $\mathcal{C}'_t$. Next, we run $k$ copies of the hallucination detection algorithm: the $i$-th copy is given as input the collection $\mathcal{L}$, the currently enumerated set $E_t$, and the target set $L_i$ as the LLM that needs to be tested for hallucinations. Our guess for the target language is the *smallest* element $z'$ for which **i)** $L_{z'} \in \mathcal{C}'_t$, and **ii)** the output of the $z'$-th copy of the hallucination detection algorithm guesses that $L_{z'}$ does not hallucinate. We now give the formal proof.

---

[5]In fact, the hallucination discards languages that are *not* subsets of $K$.

[6]In fact, we only need $2t - 1$ fresh queries in the $t$-th round.

---

**Algorithm 2** Identification via Hallucination Detection

---

**Input:** Hallucination-detection algorithm $\mathcal{D}$; enumeration $E = (w_1, w_2, \dots)$ of target language $K$; language collection $\mathcal{L}$ (with membership oracle); domain $\mathcal{X}$

1: **for** $t = 1, 2, \dots$ **do**
2:     Let $E_t = (w_1, w_2, \dots, w_t)$.
3:     Compute the *consistent set* $\mathcal{C}'_t = \{L_i \in \mathcal{L} : E_t \subseteq L_i \text{ and } i \le t\}$.
4:     **for** $i = 1, \dots, t$ **do**
5:         Run a copy of $\mathcal{D}$ with inputs $E_t$ and target set $L_i$ for $t$ steps, and obtain output $d^i_t$,
    where:
$$d^i_t = \begin{cases} 1 & \text{if no hallucinations are detected,} \\ 0 & \text{if hallucinations are detected.} \end{cases}$$
6:     **end for**
7:     Let $\mathcal{N} = \{i \le t : L_i \in \mathcal{C}'_t \text{ and } d^i_t = 1\}$.
8:     **if** $\mathcal{N} \ne \varnothing$ **then**
9:         Let $z' = \min\{i \in \mathcal{N}\}$.
10:         **return** $z'$                    ▷ Output the index of the identified language.
11:     **else**
12:         **return** 1.               ▷ We return an arbitrary index and proceed.
13:     **end if**
14: **end for**

---

*Proof of Lemma 3.2.* We let $z \in \mathbb{N}$ be the smallest number such that $L_z = K$.[7] Our algorithm outputs the language with the *smallest* index that satisfies these two conditions we described above. Thus, to get the desired result we need to show that **i)** all the languages in $\mathcal{L}_{z-1} = \{L_1, L_2, \dots, L_{z-1}\}$ that precede $L_z$ do *not* satisfy these conditions (for all sufficiently large $t$), while the target language $L_z$ *does* satisfy these conditions (again, for all sufficiently large $t$). To that end, we divide $\mathcal{L}_z$ into two disjoint subsets: $\mathcal{L}^{\supset}_{z-1} = \{L \in \mathcal{L}_{z-1} : L \supsetneq L_z\}$ and $\mathcal{L}^{\not\supset}_{z-1} = \{L \in \mathcal{L}_{z-1} : L \not\supseteq L_z\}$. In words, $\mathcal{L}^{\supset}_{z-1}$ is the set of all languages that precede $L_z$ and are strict supersets of it, and $\mathcal{L}^{\not\supset}_{z-1}$ is the set of all languages that precede $L_z$ and are *not* strict supersets of it. Notice that, since $L_z \notin \mathcal{L}_{z-1}$, we have $\mathcal{L}^{\supset}_{z-1} \cup \mathcal{L}^{\not\supset}_{z-1} = \mathcal{L}_{z-1}$. We now handle these two sets separately.

We first consider the set $\mathcal{L}^{\not\supset}_{z-1} = \{L \in \mathcal{L}_{z-1} : L \not\supseteq L_z\}$. We denote by $L_{i_1}, \dots, L_{i_k}$ the languages of this collection, where $0 \le k \le z - 1$. By definition, for every such $L_{i_j}$ there exists some element $x_{i_j} \in L_z, x_{i_j} \notin L_{i_j}$. Moreover, since the adversary presents a complete enumeration of $L_z$ there exists some timestep $t_{\ell_j}$ such that $w_{t_{\ell_j}} = x_{i_j}$ (recall that $w_{t_{\ell_j}}$ is the element enumerated by the adversary at timestep $t_{\ell_j}$.) We define $t^*_1 = \max_{j \le k} t_{\ell_j}$. Using the definition of the consistent set $\mathcal{C}'_t$ we see that for all $t \ge t^*_1$ these languages are not consistent with $E_t$, *i.e.*, $L_{i_1}, \dots, L_{i_k} \notin \mathcal{C}'_t$.

We now focus on the set $\mathcal{L}^{\supset}_{z-1}$. Let $L_{j_1}, \dots, L_{j_m}$, be the languages of this collection, where $0 \le m \le z - 1$. For any $i \le m$ consider the execution of the hallucination detection algorithm with input the collection $\mathcal{L}$, the enumeration $E$, and the target set $L_{j_i}$. Since $E$ is a valid enumeration of $L_z$ and $L_{j_i} \supsetneq L_z$, by definition of the hallucination detection property, there exists some $t'_{\ell_i}$ such that for all $t \ge t'_{\ell_i}$ the hallucination detection algorithm declares that $L_{j_i}$ hallucinates. To see that, notice that since $L_{j_i} \supsetneq L_z$ and the hallucination detection algorithm observes a sequence that enumerates all of $L_z$, it must eventually conclude that $L_{j_i}$ hallucinates. We define $t^*_2 = \max_{i \le m} t'_{\ell_i}$. It follows that for all $t \ge t^*_2$ the hallucination detection algorithm declares that each $L_{j_1}, \dots, L_{j_m}$ hallucinates.

Lastly, let us consider the language $L_z$. First, notice that for all $t \ge z$ we have $L_z \in \mathcal{C}'_t$. Moreover, using the exact similar reasoning as in the above paragraph, there exists some

---

[7]Recall that a language is allowed to appear multiple times in the collection $\mathcal{L}$.

timestep $t'$ such that for all $t \geq t'$ the hallucination detection algorithm declares that $L_z$ does not hallucinate. Let $t_3^* = \max\{z, t'\}$.

We now have all the ingredients we need to prove our result. We let $t^* = \max\{t_1^*, t_2^*, t_3^*\}$. Consider any $t \geq t^*$. By definition of $t^*$, for any language $L_i, i < z$, either $L_i \notin \mathcal{C}'_t$ or the hallucination detection algorithm with input set $L_i$ declares that $L_i$ hallucinates, so the two conditions are not simultaneously satisfied for this language. However, both conditions are simultaneously satisfied for $L_z$ since $L_z \in \mathcal{C}'_t$ and the hallucination detection algorithm declares that $L_z$ does not hallucinate. Hence, the smallest indexed language that satisfies both of our conditions is indeed $L_z$. Consequently, our algorithm achieves identification in the limit. $\square$

We now note that the proof of Theorem 2.1 is an immediate corollary of Lemmas 3.1 and 3.2.

### 3.2 Proof of Theorem 2.3

Unlike Theorem 2.1, the technical details of Theorem 2.3 are not as challenging. The full proof of this result is given below.

*Proof of Theorem 2.3.* We first describe the strategy we use to achieve hallucination detection. Recall that $G$ denotes the set we test, $K$ denotes the target language, and $E = \{(w_1, y_1), (w_2, y_2), \ldots\}$, where $y_i = \mathbb{1}\{w_i \in K\}, \forall i \in \mathbb{N}$, in a labeled enumeration of $\mathcal{X}$, *i.e.*, every element $x \in \mathcal{X}$ appears at some finite position in the enumeration, and its label indicates whether it is part of the target language $K$. In every step $t = 1, 2, \ldots$, we do the following: For every element in the input stream that appears with a 0 label, *i.e.*, $(w, 0) \in E_t := \{(w_1, y_1), \ldots, (w_t, y_t)\}$ we check if $\mathbb{1}\{w \in G\} = 1$. If this holds for some $(w, 0) \in E_t$ we declare that $G$ hallucinates. Otherwise, we declare that $G$ does not hallucinate. We now prove the correctness of this strategy. Similarly as before, we divide the analysis into two cases: If $G \subseteq K$, then the above algorithm correctly declares that $G$ does not hallucinate in every step $t \in \mathbb{N}$. This is because $w \notin K \implies w \notin G$. If $G \not\subseteq K$, we consider an enumeration of the domain $\mathcal{X} = \{x_1, x_2, \ldots\}$. Let $i^* \in \mathbb{N}$ be the smallest number such that $x_{i^*} \in G$ and $x_{i^*} \notin K$. Notice that such an $i^*$ does exist. Moreover, there exists some $t^*$ such that $(w_{t^*}, y_{t^*}) = (x_{i^*}, 0)$. Thus, for this tuple we get $y_{t^*} = 0$, and $\mathbb{1}\{w_{t^*} \in G\} = 1$. Hence, for any $t \geq t^*$ our algorithm will correctly declare that $G$ hallucinates. This concludes the proof. $\square$

## 4 Conclusion

In this work, we initiated the formal study of automated hallucination detection by introducing a mathematical framework to explore the possibilities and inherent limitations of this task. Our results provide theoretical justification for several phenomena observed experimentally. Specifically, we showed that hallucination detection is typically unattainable if detectors are trained solely on *positive* examples from the target language (*i.e.*, factually correct statements). In stark contrast, when detectors have access to explicitly labeled *negative* examples—factually incorrect statements—hallucination detection becomes tractable for all countable collections. These findings underscore the critical role of human feedback in practical LLM training. Several compelling directions for future work remain open. It would be valuable to quantify precisely the amount of negative examples needed for reliable hallucination detection, and formally explore the computational complexity of the detection problem within our proposed framework. Additionally, investigating whether hallucination detection remains tractable under noisy negative examples, as well as exploring alternative forms of feedback beyond explicit labeling, are promising avenues that warrant further exploration. Finally, inspired by the definition of Kleinberg & Wei (2025) it would be interesting to explore whether we can achieve a more relaxed notion of hallucination detection, where we only wish to detect whether the "frequency" of hallucinations is greater than some target threshold $c > 0$.

## Acknowledgments

Amin Karbasi acknowledges funding in direct support of this work from NSF (IIS-1845032), ONR (N00014- 19-1- 2406), and the AI Institute for Learning-Enabled Optimization at Scale (TILOS). Grigoris Velegkas was supported in part by the AI Institute for Learning-Enabled Optimization at Scale (TILOS)

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

# A  Preliminaries

Building on the foundational work in learning theory by Gold (1967) and Angluin (1988), Kleinberg & Mullainathan (2024) introduced a rigorous framework for language generation. In this model, the domain $\mathcal{X}$ is a countable set, and the target language $K$ is an unknown subset of $\mathcal{X}$.

## A.1  Language Identification in the Limit

The problem of language identification in the limit from positive examples was introduced by Gold (1967) and further studied by Angluin (1979; 1980). For a fixed collection $\mathcal{L}$, an adversary and an identifier play the following game: The adversary chooses a language $K$ from $\mathcal{L}$ without revealing it to the identifier, and it begins *enumerating* the strings of $K$ (potentially with repetitions) $w_1, w_2, \ldots$ over a sequence of time steps $t = 1, 2, 3, \ldots$. The adversary can repeat strings in its enumeration, but the crucial point is that for every string $x \in K$, there must be at least one time step $t$ at which it appears. At each time $t$, the identification algorithm $I$, given the previous examples $w_1, w_2, \ldots, w_t$, outputs an index $i_t$ that corresponds to its guess for the index of the true language $K$. Language identification in the limit is then defined as follows.

**Definition 3** (Language Identification in the Limit (Gold, 1967)). *Fix some $K$ from the language collection $\mathcal{L} = \{L_1, L_2, \ldots\}$. The identification algorithm $I = (I_t)$ identifies $K$ in the limit if there is some $t^* \in \mathbb{N}$ such that for all steps $t > t^*$, the identifier's guess $i_t$ satisfies $i_t = i_{t-1}$ and $L_{i_t} = K$. The language collection $\mathcal{L}$ is identifiable in the limit if there is an identifier that identifies in the limit any $K \in \mathcal{L}$, for any enumeration of $K$. In this case, we say that the identifier identifies the collection $\mathcal{L}$ in the limit.*

Angluin's seminal result (Angluin, 1980) proposed a condition that precisely characterizes which collections are identifiable in the limit.

**Definition 4** (Angluin's Condition (Angluin, 1980)). *Fix a language collection $\mathcal{L} = \{L_1, L_2, \ldots\}$. The collection $\mathcal{L}$ is said to satisfy Angluin's condition if for any index $i$, there is a tell-tale, i.e., a finite set of strings $T_i$ such that $T_i$ is a subset of $L_i$, i.e., $T_i \subseteq L_i$, and the following holds:*

$$\text{For all } j \geq 1, \text{ if } L_j \supseteq T_i, \text{ then } L_j \text{ is not a proper subset of } L_i.$$

*Further, the tell-tale oracle is a primitive that, given an index $i$, outputs an enumeration of the set $T_i$.*

Formally, Angluin (1980) showed the following result.

**Theorem A.1** (Characterization of Identification in the Limit (Angluin, 1980)). *A language collection $\mathcal{L}$ is identifiable in the limit if and only if it satisfies Angluin's condition.*

Perhaps surprisingly, this result shows that language identification is impossible even for simple collections.

## A.2  Language Generation in the Limit

Using the same game-theoretic setting as Gold (1967), Kleinberg & Mullainathan (2024) proposed a modification of this game where the objective of the learner is to *generate* unseen elements of $K$ instead of guessing its index.

**Definition 5** (Language Generation in the Limit (Kleinberg & Mullainathan, 2024)). *Fix some $K$ from the language collection $\mathcal{L} = \{L_1, L_2, \ldots\}$ and a generating algorithm $G = (G_t)$. At each step $t$, let $E_t \subseteq K$ be the set of all strings that the algorithm $G$ has seen so far. $G$ must output a string $w_t \notin E_t$ (its guess for an unseen string in $K$). The algorithm $G$ is said to generate from $K$ in the limit if, for enumerations of $K$, there is some $t^* \in \mathbb{N}$ such that for all steps $t \geq t^*$, the algorithm's guess $w_t$ belongs to $K \setminus E_t$ (or $K \setminus E_t$ is empty). The collection $\mathcal{L}$ allows for generation in the limit if there is an algorithm $G$ that, for any target $K \in \mathcal{L}$, generates from $K$ in the limit.*

Note that for the problem of language generation to be interesting, the languages of the collection $\mathcal{L}$ must be of infinite cardinality. The main result of Kleinberg & Mullainathan

(2024) is that language generation in the limit is possible for all countable collections of languages.

**Theorem A.2** (Theorem 1 in Kleinberg & Mullainathan (2024)). *There is a generating algorithm with the property that for any countable collection of languages $\mathcal{L} = \{L_1, L_2, \dots\}$, any target language $K \in \mathcal{L}$, and any enumeration of $K$, the algorithm generates from $K$ in the limit.*

