# OpenReview forum: "(Im)possibility of Automated Hallucination Detection in Large Language Models"
_colmweb.org/COLM/2025/Conference — COLM 2025_

### Official Review · Reviewer_zwtc · 2025-05-12

**Rating:** 5
**Confidence:** 4
**Ethics Flag:** 1

**Summary:**

This paper formalizes the task of hallucination detection in LLMs by framing it as a language identification problem. It proves that reliable detection is impossible with only positive (correct) examples, but becomes feasible when negative (incorrect) examples are available. The results are grounded in classical learning theory and offer a theoretical justification for the importance of feedback-based methods like RLHF.

**Reasons To Accept:**

- This paper provides a novel theoretical framing of hallucination detection via language identification.

- The authors prove formally that negative examples are necessary for reliable detection, aligning with empirical trends.

- The paper is generally well-written.

**Reasons To Reject:**

- The paper lacks any experiments or simulations to support its claims or demonstrate relevance to actual LLM behavior.

- Theoretical insights are not translated into actionable methods or guidance for real-world hallucination detection.

- The contribution is intellectually interesting but may be better suited for a theory venue.

---

> ### Author Response · Authors · 2025-06-03
>
> We would like to thank the Reviewer for their comments. We address them below:
>
>
> > The paper lacks any experiments or simulations to support its claims or demonstrate relevance to actual LLM behavior.
>
> Our main focus is to provide a theoretical framework that explains experimental observations in automated hallucination detection for LLMs. Indeed, our results align with these works which show that incorporating feedback in the hallucination detection process leads to much better performance. Thus, we view our paper as complementing these applied works by providing a rigorous mathematical explanation of this phenomenon.
>
> > Theoretical insights are not translated into actionable methods or guidance for real-world hallucination detection.
>
> The main theoretical insight is the incorporation of negative examples in the training process of real-world hallucination detectors. Our results indicate that this is a crucial component of the ability to achieve automated hallucination detection.
>
> > The contribution is intellectually interesting but may be better suited for a theory venue.
>
> Thank you for brining this up. We would like to point out that the CFP of COLM mentions: "Science of LMs: scaling laws, **fundamental limitations**, ..., **learning theory for LMs**" (https://colmweb.org/cfp.html). Moreover, in the first iteration of COLM there were accepted theoretical papers such as https://arxiv.org/abs/2402.08164 which showed a theoretical limitation of LLMs. As we explained, we view the positioning of our paper as the first theoretical framework to explain some of the phenomena observed in experimental papers. Moreover, our abstractions are motivated by a long line of influential work on language identification, and its recent adaptation to the problem of language generation.

---

> > ### Comment · Reviewer_zwtc · 2025-06-09
> >
> > Thank you for the detailed reply. I've read your responses carefully, and I appreciate the clarifications.

---

> > > ### Author Response · Authors · 2025-06-09
> > >
> > > We would like to thank you for taking the time to read our response and for increasing your score.

---

### Official Review · Reviewer_ypnG · 2025-05-13

**Rating:** 6
**Confidence:** 2
**Ethics Flag:** 1

**Summary:**

This paper proposes a theoretical framework for evaluating the feasibility of automated hallucination detection in large language models (LLMs), grounded in the classic Gold–Angluin paradigm from language identification and its extensions to language generation. It formalizes the hallucination detection task and demonstrates that detection is generally infeasible when training relies solely on positive examples. However, the analysis shows that detection becomes viable with access to both positive and negative labeled data, offering theoretical support for practical detection approaches.

**Questions To Authors:**

See weaknesses.

**Reasons To Accept:**

1. The mathematical proofs in the paper are complete.
2. The paper presents a novel insight into automated hallucination detection in LLMs.

**Reasons To Reject:**

Although the paper presents a highly novel framework and method for automated hallucination detection in LLMs and provides mathematical proofs, I have some doubts about certain details of the ideas due to the slight disorganization in the language of the paper. These doubts can be summarized as follows:
1. The main contribution of the paper appears to be the modeling of a new framework for automated hallucination detection in large language models (LLMs) by extending the classic Gold–Angluin framework from language identification and its variants in language generation. The paper uses this framework to demonstrate that, in an unsupervised and label-free manner, LLMs cannot achieve automated hallucination detection by being fed only positive examples. It also shows that this capability can be realized through positive and negative labeling based on human feedback (e.g., the yi in Definition 2.3 on line 397 of the paper). However, the reality is that, although line 45 of the paper mentions that current state-of-the-art LLMs like Claude and GPT-4 have hallucination detection capabilities far below that of humans, this does not mean that LLMs are incapable of judging whether other LLMs or themselves have generated errors in some areas of knowledge. Some recent papers have already revealed the reflective ability of LLMs to mitigate hallucinations. Therefore, disregarding the underlying proof process entirely, I am skeptical about the correctness of the conclusions drawn in the paper. In fact, the models mentioned above are not necessarily the most advanced models in the industry. Whether the premise "no language identification algorithm exists," which the proof of Theorem 2.1 relies on, still holds under the current hallucination detection capabilities of LLMs is also questionable.
2. In proving Theorem 2.1, the paper first demonstrates Lemma 3.1 and Lemma 3.2, establishing the equivalence between the existence of a hallucination detection algorithm and the existence of a language identification algorithm within the classic Gold–Angluin framework. I question whether there is a conceptual sleight of hand occurring here. Does the L and G used for the hallucination detection algorithm and within the classic Gold–Angluin framework for language identification truly represent the same things? After reading the paper carefully, I still have not found the answer I seek: detailed definitions and explanations of L and G in the real-world context of hallucination detection. Therefore, readers are left to define them based on their own understanding as specific knowledge or semantic spaces, which may or may not align with the definitions mentioned in the classic Gold–Angluin framework.
3. As mentioned in the previous point, the paper equates the infeasibility of automated hallucination detection in LLMs with the non-existence of a hallucination detection algorithm within the Gold–Angluin framework (where hallucination is defined in the paper as the existence of elements in the generated G that are outside the language K), without providing a detailed explanation of how they are equivalent. This involves detailed interpretations of the abstract concepts L, G, K, and so on, in the context of hallucination detection. Since the paper lacks explanations on this crucial point, I believe there is still a significant gap between the paper's logic and its conclusions. This also makes the overall content of the paper rather abstract. This is not directly related to the strong theoretical nature of the research content or the focus on mathematics, but rather a result of the paper's complete omission of relevant descriptive details.
4. The paper's second contribution is to demonstrate that hallucination detection algorithms can be achieved through positive and negative labeling based on human feedback. However, as I understand the paper, it seems that these labels directly point to whether something belongs to a certain language or not (for example, the y_i in Definition 2.3 on line 397 of the paper). This appears to directly replace the condition for the existence of a language identification algorithm in the proof of Theorem 2.1 with a human conclusion about whether something belongs to K (whether there is a hallucination). So, what is the point of using an LLM to automatically detect hallucinations? The previously mentioned weakness of LLMs in detecting hallucinations was in the context of directly judging the text. If humans have already labeled parts of the text as incorrect, LLMs should be fully capable of recognizing that the generation is erroneous. If the authors cannot further elaborate on the real-world correspondences of the concepts mentioned in the conclusions of Theorem 2.3 (and how they correspond to RLHF), this conclusion seems entirely meaningless to me.

---

> ### Author Response · Authors · 2025-06-03
>
> We would like to thank the Reviewer for their comments and for finding our framework highly novel. Below we address the points they raised.
>
> > The main contribution of the paper appears to be the modeling of a new framework for automated hallucination...
>
> This is very good point to clarify. Our result shows that hallucination detection is as hard as language identification under the precise mathematical framework that we have defined. The (vast majority of) theoretical works provide *worst-case* guarantees, meaning that the results hold under the assumption that nature behaves in a worst-case way for the algorithm. For example, while most of the interesting optimization problems are NP-hard, there are many practical heuristics that work well on a wide range of instances. Nevertheless, problems that admit a theoretically polynomial-time algorithm tend to admit much faster practical algorithms too, so a theoretical understanding is still useful for practical applications. Our results also have a similar flavor: when there are no negative examples, in theory there are no detectors with good worst-case performance (think of NP hard problems), and when there are negative examples detection is easy (think of problems like shortest path). This is corroborated by applied works which show that when there is no feedback hallucination detection is much harder compared to having feedback. To be clear, our results do *not* say that hallucination detection rate in practice (without negative examples) should be 0%, they say that this rate cannot be (anywhere close to) 100%. Hence, our results indeed align with practice.
>
> > In proving Theorem 2.1..
>
> Thank you for reading the paper carefully. We have provided an example (Lines 178-182) to instantiate the framework. The definitions are, indeed, the same as the Gold-Angluin framework and its recent adaptation by Kleinberg and Mullainathan: $\mathcal{L}$ is the set of languages (for instance, it could be the set of regular languages), $K$ is the *target* language (the language which specifies exactly what is true -- everything in $K$ is true, everything outside of $K$ is incorrect), and $G$ is the set of strings outputted by the LLM that the detector tests. The detector gets a stream of examples from $K$ and has access to $G$ (it can ask if $x \in G$ for strings $x$). Moreover, the detector knows that $K \in \mathcal{L}$. We have explained the hallucination detection game in line 166-176.
>
> > As mentioned in the previous point...
>
> Please see our previous answer. We would appreciate it if you could let us know whether things are clear now. We're also happy to elaborate further.
>
> > The paper's second contribution is  ...
>
> The main point of this result is to highlight that the conclusion about the feasibility of hallucination detection changes when feedback is incorporated in the training process of the detector. In the current version of the statement, feedback comes in the former of negative examples. This isn't the only type of feedback the algorithm can use in practice, it's merely used to illustrate our results within the theoretical abstraction. Other types of feedback include, e.g., the ability to ask an oracle whether something is correct or not (for a limited amount of times). Such oracles are already implemented in practice, e.g. in the form of RAGs or proof verifiers such as Lean. Moreover, the idea of the result is that by seeing a (finite) amount of labeled data the hallucinator detector is able to identify $K$ and then use this knowledge to implement hallucination detection. Therefore, it isn't as straightforward as directly asking a human annotator for the label of *every* single output of the LLM. Please let us know if you need further clarifications on this result.

---

> > ### Comment · Reviewer_ypnG · 2025-06-07
> >
> > Thank you very much for the authors' response. After carefully reading the authors' response and the comments from other reviewers, I still feel somewhat confused about certain core points and the overall purpose of the study.
> >
> >
> > 1. Regarding the claim made by the authors that hallucination detection is as hard as language identification — I have no issue with the logic of the proof itself. However, under such broad and abstract definitions, I could arguably use the classic Gold–Angluin framework to show that most theoretical problems are not solvable in the worst-case scenario (i.e., the detection rate cannot be anywhere close to 100%, as the authors mentioned). In fact, all I need to do is define a concept without providing explicit algorithms or step-by-step procedures — something entirely feasible. The authors' analysis of "hallucination detection" as a concrete theoretical problem is defined in a highly abstract manner, with little connection to practical scenarios. In their response, the authors gave an example; perhaps they can elaborate more clearly how K, L, G correspond concretely, and explicitly link the hallucination detection process to the "hallucination detection game" described in lines 166–176, so that I can better understand the relationship between the impossibility of automatic hallucination detection in the worst case and the theorem being proved.
> > 2. As briefly mentioned in my first point, the paper seems somewhat disconnected from the latest developments in the academic community — many of the cited works appear outdated. This may lead readers to question whether the paper has kept up with recent advances in the field, and even cast doubt on the validity of the paper's conclusions.
> > 3. As I mentioned in my fourth concern regarding the definition of negative samples: the authors stated that feedback does not necessarily come from humans, and there already exist methods for determining positive and negative examples for certain tasks (e.g., RAG-based approaches or tools like Lean in formal verification). I am glad that the authors provided specific examples here, which helped me better understand the paper. However, I cannot help but question the motivation behind this contribution: if we already have tools that can determine correctness, what is the point of using an LLM to do so?

---

> > > ### Author Response · Authors · 2025-06-08
> > >
> > > We would like to sincerely thank the Reviewer for taking the time to read our rebuttal and the comments, engage with us during the discussion period, and providing feedback to make our paper more accessible to a broader audience.
> > >
> > > > Regarding the claim  ... being proved.
> > >
> > > We would like to clarify that in the Gold-Angluin framework it is *not* the case that all problems are hard. In fact, as we mention in Lines 202-204, the recent work of Kleinberg and Mullainathan showed that language generation is a *provably* easy problem under the Gold-Angluin framework since it can be achieved for all collections of languages. This was precisely the starting point of our work: our goal was to understand if hallucination detection is as hard as language identification or as easy as language generation (or somewhere in between). We are happy to elaborate on the discussion of $\mathcal{L}, K, G.$ One concrete instantiation is to choose $\mathcal{L}$ to be the set of all regular languages defined over a binary alphabet $\{0,1\}$; these are the "easiest" languages in the Chomsky hierarchy and clearly much less complicated than English. The set $K$ is a regular language, e.g., a concrete instantiation would be the language of all finite strings that have the same number of 0,1. Thus, under this instantiation, we say that all strings that do *not* contain the same number of 0,1 are *hallucinations.* The set $G$ represents the (infinite) set of responses the LLM can give. We assume that the hallucination detection algorithm has *membership query* access to $G$; it can ask for any $x$ if $x \in G$ (i.e., if this string can be generated by the LLM). In practice, this can be achieved by looking at the log-probabilities of the next token. Now the hallucination detection algorithm observes a stream of strings from $K$ (without having an explicit description of $K$, just the promise that $K$ is regular), asks membership queries to the LLM and needs to figure out whether the LLM outputs any elements that are not in $K$ (in the running example it needs to understand if $G$ contains any strings that do not have the same number of 0,1.) We don't place any constraint on the running time / architecture of the detector, we just want it to achieve this in some *finite* time. Our result says given any hallucination detection strategy, we can always find some regular language for which it fails in this task. We view this as a rather strong negative result: it applies even for regular languages, under strong access to the LLM, without placing any computational constraints on the detector, and assuming that there is no noise in the input stream. We are not sure what you mean with the comment "In fact, all I need to do is define a concept without providing explicit algorithms or step-by-step procedures — something entirely feasible"; if our response is still unclear please let us know.
> > >
> > > > As briefly mentioned ... paper's conclusions.
> > >
> > > We are more than happy to discuss more experimental papers in the direction of automated hallucination detection. As we mentioned in the rest of our rebuttal response, we will certainly discuss the experimental papers that other Reviewers brought to our attention. If you have any suggestions for related experimental papers that we should discuss, please list them in your comment and we will make sure to include them in the next version of our work. That said, we believe we have done a thorough overview of the *theoretical* works that are most closely related to our model. Moreover, our understanding, and please let us know if you believe otherwise, is that the state-of-the-art of the experimental papers shows that automated hallucination detection is, in general, hard, and using feedback in the form of the methods we have mentioned in our responses does, indeed, improve the hallucination detection rate. Thus, we believe our conclusion is aligned with the results of experimental papers. Please let us know if you disagree with this statement.
> > >
> > > > As I mentioned ... LLMs to do so?
> > >
> > > We are very happy to see that our first response helped you understand the contribution better, and we are happy to elaborate on it. The main idea is that this framework provides a *unified* way to use negative examples: they could come from human annotators, RAGs, theorem provers etc, and in an extension of our model they might even be noisy; the algorithm is agnostic to the way the examples were generated.
> > > Conceptually, this is similar to training a classifier on a (small set) of examples with the hope of generalizing to new ones. Similarly, the hallucination detection scheme uses a finite number of positive and negative examples and its main goal is to generalize to detecting hallucinations on unseen examples. At a technical level, in our running example from before, using a finite number of negative examples, the algorithm identifies the target language $K$, so then it learns that all valid strings should contain the same number of 0, 1.

---

### Official Review · Reviewer_TJ7A · 2025-05-16

**Rating:** 5
**Confidence:** 5
**Ethics Flag:** 1

**Summary:**

This paper develops a rigorous theoretical framework to address a core open question in the safety and reliability of LLMs: is it fundamentally possible to automatically detect hallucinations without external supervision? Drawing on foundational work in learning theory, particularly Gold’s language identification and Angluin’s characterization of identifiability in the limit, the authors establish a key impossibility result: hallucination detection is equivalent in difficulty to language identification in the limit. This implies that for most practical settings, hallucination detection is not possible using only positive examples (i.e., correct statements). However, the paper also presents a positive result showing that when detectors are trained with negative examples (i.e., explicitly incorrect statements), hallucination detection becomes theoretically tractable. The work ties together deep theoretical insights with empirical observations and offers a valuable perspective on the design and limitations of current hallucination mitigation techniques such as RLHF.

**Questions To Authors:**

- You mention that extending to prompted generation is a future direction. Could you speculate on the technical hurdles in doing so, and whether the current impossibility result would still apply in that context?

- Do you envision that your theoretical results can help guide future architectures or training protocols? For instance, could your framework inspire new methods of automated feedback generation?

- Your results assume clean positive and negative examples. How sensitive is the hallucination detection guarantee to noisy or adversarially labeled training data?

- Would it be possible to empirically simulate a version of your theoretical framework (e.g., using synthetic languages) to validate your theorems numerically?

**Reasons To Accept:**

`Strong theoretical grounding:` Establishes a rigorous equivalence between hallucination detection and language identification.

`Practical relevance:` The impossibility result explains real-world challenges in hallucination detection; the possibility result justifies existing RLHF-style methods.

`Bridges theory and practice:` The paper draws meaningful connections between formal learning theory and empirical LLM behavior. The proofs are sound, and the writing is careful and thoughtful.

**Reasons To Reject:**

`Accessibility:` The paper may be difficult to fully appreciate for those unfamiliar with formal models of learning, particularly Gold’s and Angluin’s frameworks.

`Limited empirical validation:` While the paper provides theoretical justification for existing empirical findings, it does not directly verify its model's assumptions with empirical data or ablations.

`Simplified model assumptions:` The adversarial and promptless setting is appropriate for theoretical clarity but limits direct applicability to real-world LLM deployments.

`Insufficient coverage of related work:` Overlooks relevant prior research on automated hallucination detection - FACTOID: FACtual enTailment fOr hallucInation Detection - https://aclanthology.org/2025.trustnlp-main.38.pdf

---

> ### Author Response · Authors · 2025-06-03
>
> We would like to thank the reviewer for their thoughtful comments. Below we respond to the points they have raised.
>
> > The paper may be difficult to fully ... frameworks.
>
> We have tried to make our results accessible to a broad audience that might not be familiar with the Gold-Angluin setting by explaining their informal versions in the Introduction. However, taking the reviewers feedback into account we will make further edits in the next version: we will have explicit informal theorem statements in the Introduction that state the conceptual takeaways of our results, we will further emphasize how our results can inform the design of automated detection methods (see below), and we will elaborate even further on the connections with experimental work. We are very happy to make further edits to make our paper more accessible, if the Reviewer has concrete suggestions.
>
> > While the paper provides theoretical justification..
>
> Our model is grounded on a theoretical framework that has been fundamental to Learning Theory and Computational Linguistics, and has gained attention recently due to its adaptation to Language Generation by Kleinberg and Mullainathan. Moreover, we have tried to explain its connection to practical applications in Lines 205-247. If the Reviewer has concrete criticisms about the theoretical model, we would be very happy to try to address them and improve it. Similarly, if the Reviewer has some concrete suggestions for experiments that would help verify the model's assumptions we would also be happy to try to implement them. Moreover, we would like to point out that the CFP of COLM lists papers that show limitation of LLMs and develop theory of LLMs as topics of interest. Indeed, in the first iteration of COLM there were accepted theoretical papers such as https://arxiv.org/abs/2402.08164 which showed a theoretical limitation of LLMs.
>
> > Simplified model assumptions..
>
> These are very important points to clarify. Regarding the *promptless* setting, our main technical result is an impossibility one. Hence, an impossibility result for an easier setting (the promptless one) immediately implies an impossibility result for a more involved one (the prompted setting). We will explain this in the next version of our work. Regarding the *adversarial* setting, while this might indeed seem like it makes the problem harder, building on technical tools developed by Kalavasis et al. (2025), we can extend it to a *statistical* setting where there is a distribution over $K$ and the LLM (and the hallucination detector) are trained on i.i.d. samples from the distribution. We have mentioned it in lines 216-222, and we will elaborate further in the next version.
>
> > Insufficient coverage of related work... FACTOID: FACtual enTailment fOr hallucInation Detection - https://aclanthology.org/2025.trustnlp-main.38.pdf
>
> Thank you for bringing it to our attention, we will discuss it in the next version. We would like to point out that our main focus is developing a *theoretical* framework, and, to the best of our knowledge, such a framework does not exist in prior work.
>
> > You mention that extending to prompted generation ...
>
> Our impossibility result directly translates to the prompted setting. The positive result, which shows possibility of detection using positive and negative examples also applies when such examples are available for every prompt separately. The main technical hurdle would show up in establishing a positive result which leverages labeled data from some prompt $p$ to decide hallucination under prompt $p'$.
>
> > Do you envision that your theoretical results can help guide...
>
> This is a great point. We would our results will further highlight the necessity of feedback in the training process. While for some tasks obtaining feedback might be costly, in other areas such as theorem proving, such feedback is easier to obtain using verifiers such as Lean. Therefore, we would like to see how statements, such as explicitly labeled incorrect proofs, can be further used in training protocols.
>
> > Your results assume clean positive and negative examples...
>
> This is also a great question. Our techniques can show a guarantee of the following form: if there are $m$ corrupted examples, then the detector can detect hallucinations up to some accuracy $O(k)$.
>
> > Would... validate your theorems numerically?
>
> While something like that could definitely be possible, we aren't sure how it would add value to the paper. Our theoretical results provably hold within the theoretical model, so if we create a set of languages that fall within this model the numerical results will validate our findings. What we believe is more interesting is the fact that our theoretical results under an *abstracted* model validate the experimental results under *real-world* models. If the Reviewer sees any value to including such synthetic experiments we would be happy to revisit.

---

### Official Review · Reviewer_T4oz · 2025-05-24

**Rating:** 6
**Confidence:** 3
**Ethics Flag:** 1

**Summary:**

The paper demonstrates a theoretical framework to analyze automatic hallucination detection in LLMs by adapting Gold-Angluin language identification framework. The authors provide two fundamental results: First, hallucination detection is equivalent, in terms of difficulty, to language identification, which is known to be impossible for most languages due to Angluin's characterization. Second, they show that introducing negative examples (incorrect statements with labels) changes the landscape, making hallucination detection possible.

The model formalizes hallucination detection as determining whether an LLM's output set G is contained within a target language K (correct statements).

**Questions To Authors:**

What do you define as "hallucination"? Borrowing terms from [2], is it extrinsic/intrinsic hallucination or factuality? I don't believe it was clearly defined in the paper.

Do your results apply if the detector is implemented via an LLM — for instance, if one uses a naive pretrained model such as Llama-3.1 8B Base, or an instruction-tuned model such as GPT4o, Llama-3.1 8B Instruct acting as a verifier for another model's answer using zero-shot prompting? I want an explanation on both situations, since the former case is only trained on large text corpora, without data specifically labeled as "incorrect examples". Also, I believe even instruction-tuned models aren't likely to be trained with data that you define as "incorrect examples" (explicitly labeled incorrect statements). Of course, they would have gone through post-training process such as SFT, RLHF or DPO, but I don't think the widely-used negative samples used in such process (e.g., negative examples in preference pairs used for such process in datasets like [3]) can account for what you state as "incorrect examples". So based on my understanding, your results should apply even when LLMs are used as hallucination detectors, but if that's not the case, please clarify further.

[2] Bang, Yejin, et al. "HalluLens: LLM Hallucination Benchmark." arXiv preprint arXiv:2504.17550 (2025).
[3] Cui, Ganqu, et al. "ULTRAFEEDBACK: Boosting Language Models with Scaled AI Feedback." International Conference on Machine Learning. PMLR, 2024.

**Reasons To Accept:**

The paper provides a well-formulated theoretical result grounded in classical learning theory, bringing formal tools to the hallucination detection problem.

The equivalence between hallucination detection and language identification in the limit is cleanly presented.

The paper offers conceptual insight into why human feedback may be necessary for effective hallucination detection.

**Reasons To Reject:**

The authors' definition of hallucination is unclear. For a better understanding of their argument, the authors should provide a clear definition of hallucination.

I believe authors should mention and engage with recent empirical works such as [1], which demonstrates that LLMs are capable of detecting hallucinations without training on explicit hallucination labels.

No concrete claims are made about how this result should inform the design or evaluation of current LLM systems. If their results do not apply to LLMs, which are widely adopted as hallucination detectors in most works, it’s unclear what the practical implications are.

While the theoretical contribution is clear, the paper does not provide any empirical experiments to validate its abstractions or show how the theory predicts/model observed behavior in real LLMs. Even a toy experiment or simulation would strengthen the impact.

[1] Sriramanan, Gaurang, et al. "Llm-check: Investigating detection of hallucinations in large language models." Advances in Neural Information Processing Systems 37 (2024): 34188-34216.

---

> ### Author Response · Authors · 2025-06-03
>
> We would like to thank the reviewer for their thoughtful comments and suggestions. We provide answers to their comments below:
>
> > The authors' definition of hallucination is unclear. For a better understanding of their argument, the authors should provide a clear definition of hallucination.
>
> We provide the definition of hallucinations in Lines 166-176. Since we work in the promptless model, we identify an LLM with the set of responses it can produce, denoted as $G$ in our paper. We assume there is some target language $K$, which is again a set. If the set $G$ contains any elements outside of $K$ we say that the LLM hallucinates. The interpretation is that $K$ contains all the factually correct statements, and everything outside of $K$ is incorrect. Even under this simplified setting we show that automated hallucination detection is hard.
>
> > I believe authors should mention and engage with recent empirical works such as [1], which demonstrates that LLMs are capable of detecting hallucinations without training on explicit hallucination labels.
>
> Thank you very much for pointing out this empirical work, we will certainly discuss in the next version of our paper. The main goal of our work is to introduce the first *theoretical* framework of automated hallucination detection in order to rigorously argue the capabilities and limitations of detectors. We view our results as theoretically supporting experimental findings, such as the ones in [1]. In particular, this paper shows that the use of RAG systems improves the hallucination detection rate. In the abstraction we have proposed, the use of such systems can be thought of having access to negative examples. Therefore, we view our results as complementing the results of experimental papers from a theoretical point of view. We will elaborate more in the next version of our work.
>
> > No concrete claims are made about how this result should inform the design or evaluation of current LLM systems. If their results do not apply to LLMs, which are widely adopted as hallucination detectors in most works, it’s unclear what the practical implications are.
>
> Our results apply to any automated hallucination detection system, therefore they also apply to LLMs as hallucination detectors. The main takeaway message that we hope can inform the design of current LLM systems is the incorporation of negative examples in the training process / hallucination detection process, especially in applications where hallucinations can be very damaging. These negative examples can be provided, implicitly, by external systems such as RAGs. In other applications, such as reasoning models for scientific problems, systems that give negative examples can be implemented by automated proof checkers such as Lean. We will mention this, and related takeaways, in the next version of our work.
>
> > While the theoretical contribution is clear, the paper does not provide any empirical experiments to validate its abstractions or show how the theory predicts/model observed behavior in real LLMs. Even a toy experiment or simulation would strengthen the impact.
>
> Thank you for brining this up. As we explained, we view the positioning of our paper as the first theoretical framework to explain some of the phenomena observed in experimental papers such as [1]. Moreover, our abstractions are motivated by a long line of influential work on language identification, and its recent adaptation to the problem of language generation. In terms of providing experiments, for the impossibility result of automated hallucination detection, we aren't sure what experiments would be appropriate since this is a largely negative result. Lastly, we would like to point out that the CFP of COLM mentions: "Science of LMs: scaling laws, **fundamental limitations**, ..., **learning theory for LMs**" (https://colmweb.org/cfp.html). Moreover, in the first iteration of COLM there were accepted theoretical papers such as https://arxiv.org/abs/2402.08164 which showed a theoretical limitation of LLMs.
>
> > What do you define as "hallucination"? Borrowing terms from [2], is it extrinsic/intrinsic hallucination or factuality? I don't believe it was clearly defined in the paper.
>
> Please see our previous comment. In the terminology of [2], our definition is closer to factuality: there is some target language $K$ that contains all the factually correct statements, at any given point the LLM observes training data from $K$ and we measure whether what it outputs is outside of $K$ or not.
>
> > Do your results apply... when LLMs are used as hallucination detectors, but if that's not the case, please clarify further.
>
> Your understanding is correct; our results apply even when LLMs are used as hallucination detectors. While our theoretical result requires explicitly incorrect statements, it is possible to show that other type of feedback, such as the one obtained in RLHF, improves hallucination detection. We will elaborate in the next version.

---

> > ### Comment · Reviewer_T4oz · 2025-06-03
> >
> > First, I agree with the authors that theoretical papers should indeed be counted as meaningful contributions and be presented at COLM.
> >
> > However, your response is still somewhat confusing.
> >
> > - Your understanding is correct; our results apply even when LLMs are used as hallucination detectors. While our theoretical result requires explicitly incorrect statements, it is possible to show that other type of feedback, such as the one obtained in RLHF, improves hallucination detection. We will elaborate in the next version.
> >
> > What do you mean by this? What I am asking is, based on my understanding of your claim, LLMs such as LLama-3.1 8B Base (which has only been pretrained on large text corpora), or Llama-3.1 8B Instruct (which has been post-trained, but I believe has never been trained with a "negative example" of your definition) should **not** be able to detect hallucination automatically, since they have not been explicitly trained with negative examples of hallucination. Rather, most post-training datasets are just preference datasets, just as I have cited in my original review. Such datasets shouldn't be counted as negative **hallucinated** examples, so models that have been instruction tuned (GPT-4o, Claude 4 Sonnet, Gemini, etc) shouldn't be able to detect hallucination automatically. Is this what the authors are arguing? Please clarify further.
> >
> > While I do not modify my score instantly, I will lower my confidence and am willing to raise my score based on a further detailed explanation of the application to LLMs. If the other reviewers raise the score, I will agree with them and lean towards acceptance.

---

> > > ### Author Response · Authors · 2025-06-05
> > >
> > > Thank you for your quick response and for recognizing the contributions of theoretical works to the COLM community.
> > >
> > > Regarding your question, our result does not imply that LLMs trained only on positive data can *never* detect hallucinations, but rather that LLMs trained on positive data cannot detect hallucinations at a high rate. For instance,
> > > in the LLM-Check paper you have cited, the detection rate is between 50-70%, so we still view it as in line with the theoretical results we have shown.
> > >
> > > As a more general comment, we would like to point out that the (vast majority of) theoretical works provide worst-case guarantees, meaning that the results hold under the assumption that nature behaves in a worst-case way for the algorithm. For example, while most of the interesting optimization problems are NP-hard, there are many practical heuristics that work well on a wide range of instances. Nevertheless, problems that admit a theoretically polynomial-time algorithm tend to admit much faster practical algorithms too, so a theoretical understanding is still useful for practical applications. Our results also have a similar flavor: when there are no negative examples, in theory there are no detectors with good worst-case performance (think of NP hard problems), and when there are negative examples detection is easy (think of problems like shortest path). This is corroborated by applied works which show that when there is no feedback hallucination detection is much harder compared to having feedback. To be clear, our results do not say that hallucination detection rate in practice (without negative examples) should be 0%, they say that this rate cannot be (anywhere close to) 100%. Hence, our results indeed align with practice.
> > >
> > > We would appreciate it if you could let us know whether our response clarifies your concern.

---

> > > > ### Comment · Reviewer_T4oz · 2025-06-05
> > > >
> > > > I appreciate the authors' response. I am convinced by the authors and therefore raise my score. I would appreciate it if the authors could further add detailed explanations related to the actual LLMs. I think it would benefit the paper, considering that the current version may be difficult for researchers in the LLM field to understand clearly.
> > > >
> > > > I hope your response convinces the other reviewers as well. Good luck!

---

> > > > > ### Author Response · Authors · 2025-06-08
> > > > >
> > > > > Thank you very much for continuing to engage with us during the discussion period. We are very happy that you support our paper. We will certainly add the detailed explanations and the comments we discussed to make the paper more accessible. Our main goal of submitting our work to COLM instead of a more theoretical venue was to engage with applied researchers and make our results more accessible to this community, so we are glad that we are getting feedback to achieve that!

---

### Decision · Program_Chairs · 2025-07-08

**Decision:**

Accept

**Comment:**

The paper offers a clear, theoretically grounded answer to whether (one theoretical framing of) hallucination detection can be automated: under the classic Gold-Angluin model, detection without negative examples is as hard as language identification and thus impossible in the worst case, while even sparse labeled negatives make it tractable, conceptually explaining why feedback-rich methods like RLHF help. As is common in such theoretical results, there may exist a gap between "hallucination detection" as is empirically measured on finite benchmarks, and the theoretical claims on the most general case.

Reviewers split (6, 6, 6, 5), with concerns about abstraction, lack of experiments and practical conclusions, yet they acknowledge the validity of the proofs, a novel framing, and alignment with empirical trends. After engaging in the discussion, the lowest scores rose, and clarifications showed how the framework maps to real-world detectors (e.g., RAG supplying negatives, thus not contradicting the theoretical results). Remaining weaknesses are absence of empirical validation and accessibility to non-theorists, but the contribution fits COLM’s "fundamental limitations" scope well. Overall, I support acceptance provided the camera-ready version adds concrete examples, better framing with respect to current detection work, and a brief empirical sanity check to aid audiences that are less theoretically-leaning.